# The Impact of VR Application on Student's Competency Development: A Comparative Study of Regular and VR Engineering Classes with Similar Competency Scopes

**Jang Hee Lee and Olga A. Shvetsova \*** 

School of Industrial Management, Korea University of Technology and Education (KOREATECH), Cheonan 31253, Korea; janghleel@koreatech.ac.kr

\* Correspondence: shvetsova@koreatech.ac.kr; Tel.: +82-10-9996-9553

**Abstract:** This research focuses on the Fourth Industrial Revolution and the Korean National Competency Standards. The aim of this research is to perform a comparative study of two university engineering courses to present the difference between the effect of virtual reality (VR)-based teaching and traditional teaching on learning robotics. To achieve this goal, the authors used the expert method, focus group interviews, and a comparative study. There were three hypotheses and four research questions on the relationship between teaching method and competency development. As a case study for this research, the authors chose Korea University of Technology and Education and designed a survey to assess students' competencies. The participants were second- and third-grade university students. Quantitative data were collected from the interview forms developed by the researcher; qualitative data were analyzed using the expert method and a comparative study. A significant difference was observed between competency development in the VR class and that in a regular class with a similar syllabus. Furthermore, it was noted that different teaching methods affect competency development in different ways; VR tools develop some competencies, but not all of them. Authors found that the VR tool requires an appropriate teaching method and affects the student's ability to develop competencies.

**Keywords:** university competency model; Fourth Industrial Revolution; engineering course; VR class; regular class; comparative study

## 1. Introduction

### 1.1. VR Technologies and Tools in Education: Modern Review

Virtual reality (VR) technologies were rapidly recognized in construction engineering education and training (CEET) programs because they are believed to effectively enhance the quality of such programs. A representative taxonomy of the visualization system for classifying types of VR was originally proposed by Chee [1] and Chen et al. [2], and it describes how "virtual" and "real" are merged in different proportions to create a visualization environment. There are four different levels on the reality–virtuality (RV) continuum to be defined: pure real presence, augmented virtuality (AV), augmented reality (AR), and pure virtual presence. Strictly speaking, according to Bollinger et al. and Shin, innovation technologies help develop a competitive product; this is why, currently, VR is attracting much attention for improving communications in the professional workplace and other shared spaces [3–5].

Formulating the educational paradigm of the new industrial revolution, we proceed from the assumption that this paradigm should not conflict with the paradigm of industrial production. Tranfield

et al. confirmed that the presence of such a conflict in learning situations, for example, learning in a tough educational environment and performing additional work that requires maximum flexibility, significantly reduces the effectiveness of training [6].

VR is a model of reality created by technical means, the objects and subjects of which are perceived by humans through their sensations: sight, hearing, smell, and touch. The first attempts to create an artificial reality that was no different from true reality were made in the early 1960s and aimed to develop multi-touch simulators that could transmit the dynamics of images, sounds, and smells to the user in real time. Recently, the different ways in which VR may be used with other teaching methods (for example, problem/project-based learning, computer-assisted teaching, object-oriented programming concepts) were discussed by Calder, Birinci Konur et al., Erden et al., and Kanaki et al. [7–10]. With the development of computer technology and information technology, technological platforms and opportunities for creating virtual reality in various fields of human activity emerged.

Visvizi et al. [11] explained that, if we follow the principle of compliance with the production and educational paradigm, then, naturally, the need for systemic changes in the educational environment will follow. They mentioned that such changes are already being implemented in leading foreign universities and can be classified as follows:

- Digitalization of education, which, in essence, means expanding, through digital solutions and information systems, students' access to (1) educational resources from the best universities in the world, (2) remote data from the results of scientific experiments and research, (3) a library of engineering tasks and problems, and (4) the creation of distributed labor, research, and educational teams;
- Personalization of learning, which means deepening and developing the existing practice of building an individual educational trajectory in which it is possible to return to the branching points and acquire additional skills throughout one's lifetime. Another aspect of personalization is the consideration of the requirements and requests of not only students but also direct employers;
- The project approach is an integral component of in-depth education, and it significantly improves the efficiency of the educational process, from the first stage of identifying and understanding problems to the final stage of performing the practical work activity. The project approach is inextricably linked with adaptive education, the main component of which is training through practical activities in the market, industry, and science;
- Integration of formal and non-formal education, which actually means the erosion of the physical boundaries of the university and a shift in emphasis from the process of obtaining knowledge to its recognition and assessment, regardless of the actual place of knowledge and skill acquisition;
- The creation of creative spaces that actually play the role of integration platforms for students of various specialties, the real business sector and industry, and academic and vocational education. A prerequisite for working on such sites is the collaboration on projects initiated by a real sector of the economy [11].

The creation of inter-university networks (university hubs), in contrast to the usual networking of universities, means the creation of real sites in the form of research and educational centers with the participation and under the auspices of various universities.

Pantelidis et al. explained perspective on VR application in the global environment as a three-dimensional (3D) visualization and virtual reality and global visual collaboration [12]. Their approach creates global collaboration between virtual teams that are spread around the world and includes effective computer technology-related visualization, virtual reality, and simulation of realistic behavior to create an "effect of presence".

The most important question is how VR application affects the education process in different areas. Most researchers on this topic (McLellan, Pantelidis and Auld, and Salzman et al.) expounded on the advantages of VR technologies in engineering, medicine, and earth science fields [12–14]. There

are also some disadvantages of VR application in the education process, as discussed by Pantelidis and Auld, and Salzman et al. [13–15]. Recent research papers in this area focused on the benefits of VR applications not only in engineering and soft-skilled education but also in other skill-based fields (for example, Branson-Potts presented some examples in his publication) [16]. It is necessary to investigate the role of VR application in competency development and identify the impact of advanced teaching methods on the university competency model; for example, developing a 3D environment may lead the university to a competitive place in the industry. This discussion is based on publications by Carnahan, Gerson et al., Sorbi, and Chou et al. [17–21].

The recent trends in modern engineering education include the rapid growth of different technological developments. Such changes impact educational methods and transform students' competencies. There are some risks associated with these trends, because different technology applications may affect competency development in different ways. This motivates our current research on the impact of VR technology on competency development.

### 1.2. VR Application in Engineering Education: Literature Review

By the end of the 20th century, a trend emerged in the global economy, namely, a switch from the physical trade of goods and services to electronic commerce and payment with electronic money. This approach has obvious advantages. As an example, we can discuss these advantages in the case of using the 3D environment in education as follows:

- Lack of weight of goods [18];
- Virtuality [19];
- More choice,
- Instant movement from product to product [20];
- No storage costs [21];
- No rental costs [22].

Because of the obvious advantages, the electronic economy entered everyday life and occupies a sizeable market share, which, in Russia, for example, is estimated to be $20.4 billion. The size of the internet economy is estimated to be 2.3 trillion dollars in the same country. Information about computing education and its impact on the economy is provided by Fee et al. [22]. Electronic (digital, web, internet) economics is defined as economic activity based on digital technology.

The high rate of development of the e-economy led to the need for including state structures and regulators. E-government and e-government services, including those for businesses, began to appear. The acceleration of business interactions with government regulators has a positive effect on the economy as a whole because it saves time and monetary resources for supporting activities.

Virtual and augmented reality (VR/AR) is becoming one of the tools of the electronic economy. VR/AR technology first captured its most successful market—the market of computer games, which is several billion dollars—as reported by Freeman [23]. When businesses understand all of the possibilities of VR/AR technologies, such technologies begin to be actively introduced to the business environment. VR/AR technologies in business are used for visualizing premises, product shelves, internal and hidden structures, and movement and comparing various spaces. Increasingly more business areas are incorporating virtual and augmented reality into their activities. It is used in construction, retail, investment, finance, education, etc., as discussed by Portelanc, Kovacs, and Kot et al. [24,25].

VR/AR technologies are especially important in an educational environment. Education is increasingly moving to the internet environment and becoming more distance-oriented. It is no secret that modern education requires the construction of an effective multi-level system of continuous education. The benefits and risks of e-education are presented in Table 1.

**Table 1.** Benefits and risks of the electronic form of education at a university.

| Area of Investigation | Benefits | Risks | Examples |
|---|---|---|---|
| Infrastructure | Modernization of infrastructure | High costs | B (benefit): Modernization of infrastructure requires new equipment and modern technologies<br>R (risk): Implementation of new equipment and modern technologies requires additional costs, time for establishing settings, and personnel's adaptation |
| Innovation | Creation of innovations | Competition, quality of the service/product | B (benefit): Creation of innovations may lead universities to new forms and methods of education that can be more flexible and market-oriented. Universities may invent new strategies: franchising or licensing<br>R (risk): New innovations should be focused on market needs and industry standards; otherwise, they will not be successful. Innovations should provide relevant quality of product/service |
| Competency | Competency development | Competency assessment; matching industry standards | B (benefit): Relevant competencies create new standards and skills, develop new teaching methods, and help to achieve organizational leadership<br>R (risk): The competency model has some limitations—it must follow industry standards; it should provide relevant requirements for the development level of individual competencies |
| Brand | University's brand expansion | Intellectual property and rights risks | B (benefit): A university may expand its brand and improve its positioning strategy<br>R (risk): Electronic and online educational market is a fast-cycle market, so competition may rise very quickly; it is necessary to protect know-how and innovations and arrange intellectual property rights |
| Customization | New customers and profit; market expansion | Unstable market preferences (risk of customization); health threats | B (benefit): University expands strategy (even globally), finds new markets, becomes more attractive to new customers<br>R (risk): Customers' preferences are highly changeable, especially in different markets, so it is necessary to provide a differentiation strategy |

Source: made by authors.

Online training for company employees is becoming more common. However, not every company is ready to spend money on developing its own learning management system (LMS) platforms since it is cheaper to buy access to an already finished platform. This creates an additional market for e-learning courses for educational institutions.

In order to effectively sell and purposefully improve the product, it is necessary to clearly represent the modern student—the consumer of the service. Cobb et al. [26] reported that today's students best perceive information in a game form, do not read more than 800 characters at a time, acquire all of their information from the internet, have constant access to the internet, and even perceive art through gadgets.

However, the main point is that modern students always want to expand their opportunities, including opportunities for learning. Modern dynamics makes it easy to find tool guides, allowing individuals to quickly and fully acquire the necessary knowledge. Cordes et al., Li et al., and Thoben et al. [27–29] indicated that, according to the Fourth Industrial Revolution, the faster a company learns, the more it will be in demand in the labor market.

Kazakoff et al. [30–43] introduced definitions and concepts of different teaching technologies in education, so we tried to form a classification scheme of the technologies that are currently used in the education system (Table 2).

**Table 2.** Definitions of virtual reality/augmented reality (VR/AR) in education (literature review).

| Type of Technology | Definition | Author |
|---|---|---|
| VR | "Inducing targeted behavior in an organism by using artificial sensory stimulation, while the organism has little or no awareness of the interference" | LaValle (2017) [31] |
| | "Creates an artificial environment to inhabit" | Schechter (2015) [32] |
| | "An interactive computer simulation which transfers sensory information to a user who perceives it as substituted or augmented" | Chou, Smith; Ragan (2017) [33,34] |
| | "Offers significant opportunity in the area of simulation" | Hoffmann; Meisen, and Jeschke; Rodic (2014) [35] |
| | "Particular situation can be programmed with several variables and environments on which the student can act" | Dede, Salzman, Loftin (1996) [36] |
| AR | "Simulates artificial objects in the real environment" | Schechter; Kırbag Zengin; Yucasu (2015) [37] |
| | "Integrates digital information with real environments in which people live" | Korchagina, E.V., Shvetsova, O.A. (2019) [38] |
| | "Increases student's creativity without fear of manufacturing risks and costs" | Di Serio, Ibáñez, and Delgado Kloos; Mitchel and Scratch (2013) [39,40] |
| | "Provides social factor of sharing the experience in real time with real individuals: their classmates" | Ibáñez, Di Serio, Villarán, and Delgado Kloos; Ansari and Khan (2014) [41,42] |
| | "Sessions around health and engineering areas enable the teacher to share knowledge with students using images superimposed on the reality of their classrooms" | Boletsis and McCallum; Bong (2010) [43] |

Source: made by authors.

Most of the authors recognized AR/VR applications as tools that help students to better understand an environment, identify main problems, and test their solutions in specific cases (Table 3).

**Table 3.** University education methods and their combination with VR/AR applications.

| Education Method | VR | AR | Advantages | Disadvantages |
|---|---|---|---|---|
| General class | +/− | +/− | VR/AR tools save time, express practical examples. | More common links with general class material are required; VR/AR material should relate to the main lecture statements and be supportive. |
| Flip-learning | +/− | +/− | VR/AR can be implemented in both types of class activity: lecture or practice. | VR/AR can be used only for in-class activity (not for pre-class or after-class activities). |
| Project-based learning (PBL$_1$) | + | + | Students may use VR/AR tools to manage projects; VR/AR helps to conduct deep research. | VR/AR cannot cover all steps of project management; workload might be unequally shared among the members of the team. |
| Problem-based learning (PBL$_2$) | + | + | VR/AR can be implemented for case studies. Visualization of the problem is also possible; VR drives core questions and develops student's collaborative and investigative skills with regard to real-world problems. | Students consider the allocation of different project topics by professors as a disadvantage. In order to facilitate the transparency of the process, the professor needs to suggest a different topic per student or per team. This means that not every student in class is learning the same thing. More resources are needed for future VR/AR development and specifications. |
| Online education | +/− | +/− | VR/AR can help professors to manage tasks or cases easily without traveling; provides students with responsibility for self-control. | VR/AR needs more specifications, support materials, and clear tasks for self-education; students across the world should use the same technologies |

Source: made by authors.

The comparative study mentioned in the table above lets us understand the major benefits of VR applications in the education process. They can be combined with other advanced teaching methods (project-/problem-based learning (PBL), flip-learning, online education, etc.).

*1.3. Competency Model of University Students: Major Tasks of the Fourth Industrial Revolution and the Role of VR*

The Fourth Industrial Revolution is associated with the development of global industrial networks to which all production processes of various enterprises will be connected.

The integration of production and the sphere of intellectual technologies should reach a level that allows the manufactured product to interact with any necessary object in the global network, for example, using a digital teaching platform for students learning math. According to Weiss [44], the actors of this network are "smart products", which, with the help of sensors and communication systems, can be independently decentralized to manage their own production.

The functions of sourcing information, changing its environment, controlling the process of its creation, and network interaction with other elements of production are transferred to the product. It is no longer about the realization of the human–machine communication paradigm, declared in the era of automation, but about building an environment of machine–machine interactions to connect technical objects as effectively as the internet unites people. This will allow even the largest enterprises to increase work flexibility and productivity, which were previously regarded as the advantages of small or non-manufacturing companies.

Sensors, robots, and gadgets are now the participants in information-exchanging interactions. This phenomenon is reflected in the concept of the Internet of things (IoT), which was developed in the late 1990s at the Massachusetts Institute of Technology, but only in the last few years did it reach the stage of maturity and turned into a technology that can have a significant impact on existing technology and society, as Lezanski presented [45]. Its implementation implies two main directions for IoT, namely (1) the organization of a new type of industry in which products with sensors can manage their own production, and (2) integration into a global network of technical objects that surround people to create a common information environment for them.

The IoT solution is the defining task for the Fourth Industrial Revolution. The development of technologies that allow micro- and nanodevices to receive energy from the environment to meet their needs will be the turning point that will fundamentally change the production and use of products, as well as the technical, technological, and communication environment of humans. Hartmann and Halecker [46] predicted that new machines will have not only the possibility of communicating with people and artifacts but also independence of existence, autotrophy, and a higher degree of independence from humans.

Hossain and Muhammad [47] discussed outcomes of the industrial environment and innovation and provided a concept that forecasted the development of education in the 2000s on the basis of the ideas of integration and globalization, the principles of lifelong education, and the development of mobility and recognition of educational documents throughout the world. Currently, it is clear that these ideas are only partially implemented, while new trends in education continue to gain momentum. One of the key changes was the emergence in the educational market, including large universities, of new global players, such as information and communications technology-oriented companies that are actively promoting the idea of "fast technological learning".

Higher education is already facing new competitors, such as international online universities, which offer mass online courses that can remotely train a number of students who are unable to access classical schools.

The most accessible form of learning will be mass, economical, individual, "deserted" training (educational fast food) using new technological solutions, such as personalizing educational trajectories, artificial intelligence as a teacher, distance technologies and simulators, and the information educational environment. Costly high-intensity "living" education will be based on personal interaction with highly

qualified teachers, collective creative work, and the formation and development of teams, including training in special communities. These approaches may be helpful in different areas, such as marketing (Internet of things) or supply-chain management systems (logistics). These cases were discussed by Atzori et al., Kiel et al., and Koether [48–50]. Ongoing informatization set the direction for transiting to the formation of a postindustrial type of education, with a project charter focused on the creation of new competencies using modern teaching methods. There was a paradigm shift from knowledge- and content-centered teaching to competency-based, practice-oriented, and subject-centered teaching. In the latter case, the textbook and the teacher lose their position as the main sources of knowledge and information. The number of such cases is growing progressively faster; content is being updated and presented in multimedia-based interactive forms, and knowledge libraries in media formats are replacing classical libraries.

Different forms of VR can be used in different areas. The form depends on the market's goals and which VR technologies can be developed, improved, or transferred from other fields. According to the Fourth Industrial Revolution, universities try focusing on the competencies that are necessary for the market (or industry), so they use only the specific VR applications that help them to match required competencies. In our opinion, the benefits of such an approach are low market risks and fast-driven research and development (R&D); however, there are also some risks: a high level of competition among universities and markets with similar competencies and educational technologies.

The Internet of things and the Internet of everything, having provided each machine and each material object with its own position in the global digital space, will create a hybrid learning environment that will open up great opportunities for new interactive education methods. Johansson et al. [51] remarked that the development of brain–computer interfaces and the creation of appropriate network protocols could be the basis for a cognitive revolution in learning and a new generation of the internet—the NeuroNet or Internet 4.0.—that will involve people's bodies and minds in unified communication. In the near future, biometric devices can be included in educational practices to track student activity and physical indicators, adjust the methods and speed of training, and plan an individual educational program.

The set of qualifications in the conditions of the Fourth Industrial Revolution is determined by the nature of the equipment and production relations, as well as organizational, working, internal, and external environments. The main groups of competencies include the following:

1. Possession of intellectual communication technologies (ICT) at the advanced user level with knowledge of the specifics of using information systems in their professional field and the ability to set goals for the use of ICT in the production sector;
2. System-based thinking and the ability to perceive the totality of relationships and relations in the process of industrial production as a complex system with the ability to exert the necessary influence on its elements to achieve the desired result;
3. Effective interaction and group work with specialists from other professional fields, including internationally;
4. Project thinking and management skills of any activity as a project;
5. Deep professional knowledge in one's field based on an interdisciplinary approach and knowledge of related fields.

The Korean National Competency standards are a part of the policy to create a competency-oriented society in which the employees are awarded and promoted not on the basis of academic credentials and backgrounds but on the basis of competency. The Ministry of Employment and Labor developed the National Competency Standards (NCS) and many public institutions use the NCS since 2015 [52].

The aim of the NCS is the systematization of the knowledge, technology, and intelligence required of individuals to perform their duties in industrial settings. It was established by the Korean government according to industrial sectors and levels and means the standardization of competencies (knowledge,

technology, attitude) required to successfully perform duties in industrial settings at the national level (Development Manual for NCS, 2015), as introduced by Liao et al. [52].

The National Competency Standards in Korea help improve qualification for work in different areas through a variety of means, such as the following:

- Improving the suitability of education, training, and qualification for work;
- Improving the employability of learners (education, training);
- Creating links between education, training, and qualification (present overlapping investment);
- Promoting work-based life-long learning;
- Promoting the globalization of qualifications (compatibility).

Implementing VR in the education system provides more interactive visualization and interaction experience. Preuveneers and Ilie-Zudor [53]; Abatecola, G. et al. [54] asserted that sometimes it is important for students to conduct experiments and engage in some practical work, but sometimes this is not possible due to the high cost of equipment and reagents or the danger of some experiments. In virtual reality, such experiments can be carried out for as long as the user wishes; all students will be able to realize their own experiments and learn the material better.

There are three major partners in the education system in Korea: government, industry, and university (Figure 1). They have different tasks and responsibilities, but all of them participate in the competency model.

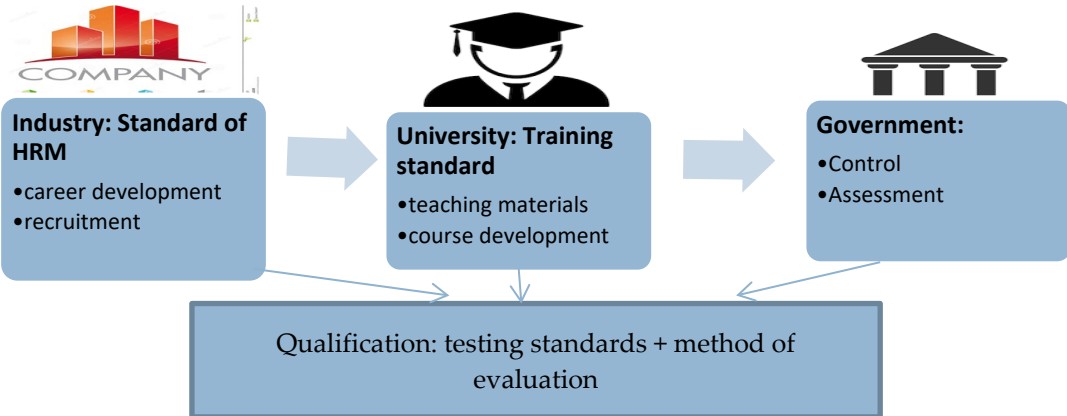

**Figure 1.** Korean model of university–industry training cooperation.

The specific feature of the Korean National Competency model is the relationship among all participants in the market and their significant roles in the creation, distribution, and development of competencies.

## 2. Research Questions and Methodology

### 2.1. Research Goals

The main research problem of this study may be recognized as the difficulty in the education process to meet the National Competency Standards within the specific tools in teaching. It means that several factors affect students' development, such as teaching method, tasks, technology tools, etc. Thus, the best way to solve this problem is to compare different approaches, investigate all factors, and find the best practical tool in the education process for appropriate competency development.

The main goal of this research is to conduct a comparative study to present the difference between VR-based teaching and traditional teaching in learning robotics. To achieve this goal, the authors created a survey of students' competencies. Therefore, three hypotheses were proposed during the development of the survey.

**Hypotheses H1.** *The VR class improves and develops core competencies in a basic engineering course better than the regular class;*

**Hypotheses H2.** *The VR class does not improve or develop core competencies in a basic engineering course better than the regular class;*

**Hypotheses H3.** *The VR class improves and develops only a certain scope of competencies in a basic engineering course.*

Four research questions were established during the development of the survey to improve the clarity of the task.

1. How can competency be developed with VR?
2. How does VR focus on competency development?
3. Which teaching methods can influence competency development in a regular class?
4. Which teaching methods affect competency development in a VR class?

*2.2. limitations of Research*

There are some limitations of this research, as follows:

- This research is based on a case study of one university (South Korea);
- The period of the survey is one semester (four months);
- The scope of this survey is focused on the competency model of the Fourth Industrial Revolution.

*2.3. Structure of Research*

The case study of Korea University of Technology and Education (hereafter referred to as KOREATECH) presents the difference between VR-based teaching and traditional teaching in learning robotics. The research team analyzed the course syllabus for "Robotics" (case study method), identified the core competencies of students (expert method), discussed teaching methods (expert method), and interviewed two groups of students (one interview for the traditional class and another for the VR class). These interviews had three stages; the first stage was performed at the beginning of the semester, the second stage was after the mid-term exam, and the final stage was at the end of the semester (Figure 2; Figure 3). A list of interview questions was prepared with the aim of showing the differences in the learning efficiency and academic performance of the student groups (comparative study method), and a retention test was conducted to compare how the students increased their knowledge and developed certain competencies.

A special VR application was developed for the "Robotics" class activity. The VR application presents a business game, which is based on a case study (problem-based story). Students were instructed to watch and listen to the story for 7–8 minutes (3–4 different problems for different groups of students), after which that they received their role. For the decision-making process, they had to play their roles and solve the problem (develop the solution) in their case study.

2.3.1. Step 1 (Figure 2)

During the first step of the research, it was necessary to analyze the course syllabus to investigate competency and teaching goals. Afterward, the teaching methods were described, and the last effort was to create a questionnaire for students' feedback.

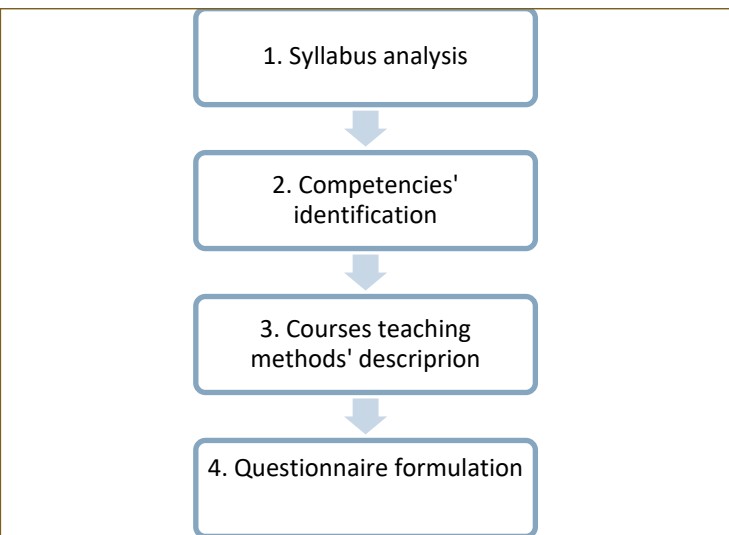

**Figure 2.** First step of research.

2.3.2. Step 2 (Figure 3)

During the second step of the research, it was necessary to arrange the focus group interview with relevant students (who attended the class); after that, with the help of individuals with expertise on competencies, the results from the interviews were compared with the answers in the channel quality indication (CQI) reports. The last effort was to analyze the results of the survey and assess the hypotheses.

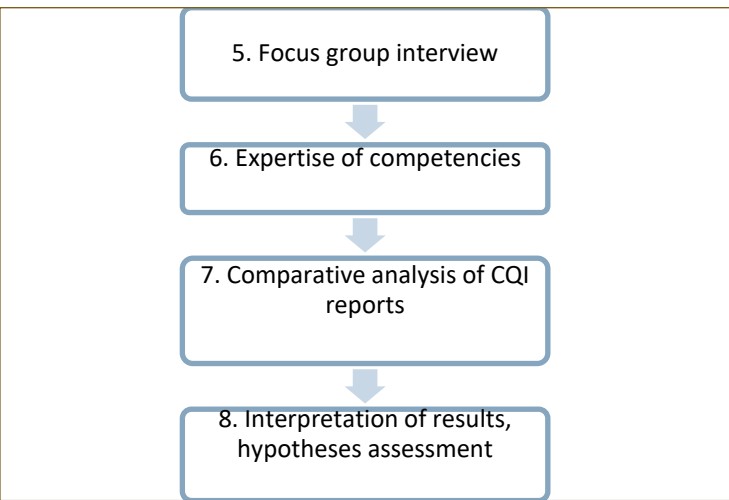

**Figure 3.** Second step of the research.

*2.4. Research Methods*

This study is based on the three following research methods:

1.    Expert method;
2.    Focus group interview;
3.    Comparative study.

The expert method helps investigate and rank all the factors; the focus group interview helps collect the data; the comparative study helps manage findings and make the decisions.

### 3. Survey and Data Collection

*3.1. Expert Method: Competency Evaluation*

For expertise, three experts from different fields were invited: one expert was from industry (company's representative person); the second expert was from the university's human resources development department (KOREATECH); the third expert was the Professor of Robotics (KOREATECH). Their tasks were to (1) identify the most important competencies and teaching methods of the "Robotics" course, and (2) estimate the impact of the teaching method on competency development in the "Robotics" course.

There are 12 competencies at KOREATECH, and they are focused on the Fourth Industrial Revolution trends, as listed below.

1. Interdisciplinary fusion;
2. Troubleshooting;
3. Communication;
4. Challenging practice;
5. Study map;
6. Global approach;
7. Major (foundation);
8. Major (specialization);
9. Practical application;
10. Positive self;
11. Human relationship;
12. Civic consciousness.

Firstly, the experts analyzed the syllabus of the "Robotics" course. An example (brief version) of the syllabus is presented in Appendix A. The syllabus identifies the goals and teaching methods of the course. The same syllabus was used for the regular class and the virtual reality class of "Robotics".

Secondly, it was necessary to compare the specified competencies to find the most important ones ("core" competencies) for the "Robotics" class.

On the basis of an expert's evaluation of competencies, students were assessed according to the class goals (in the expert's opinion) (Table 4). The goals for class "Robotics" were as follows:

- Collaborate in groups and teams;
- Design robots for specific activities and scenarios;
- Use "Robolab" programming software;
- Understand gears, pulleys, torque, friction, timing, sensors, and program loops;
- Design and develop complete robotic activities and challenges.

From this expert method, we found that the most important competencies for the course "Robotics" were troubleshooting, practical application, study map, human relationship, challenging practice, communication, interdisciplinary fusion, and major (foundation). All of these competencies had average weights of more than 3.5 points.

During the next step, it was important to check the impact of the teaching method on the competency development outcome. The experts together tried to come to a common decision (Table 5).

**Table 4.** Competency evaluation (expert method). HRD—human resources development.

| Competency | Expert 1 (Professor) | Expert 2 (HRD) | Expert 3 (Industry/Company Representative person) | Average Weight of Competency/Rank |
|---|---|---|---|---|
| Interdisciplinary fusion | 4 [1] | 3 | 4 | 3.66/7 |
| Troubleshooting | 4 | 5 | 5 | 4.66/1 |
| Communication | 4 | 5 | 3 | 4.0/6 |
| Challenging practice | 4 | 4 | 5 | 4.33/5 |
| Study map | 5 | 4 | 4 | 4.33/3 |
| Global approach | 1 | 1 | 1 | 1.0/11 |
| Major (foundation) | 4 | 3 | 4 | 3.66/8 |
| Major (specialization) | 1 | 1 | 1 | 1.0/12 |
| Practical application | 5 | 4 | 5 | 4.66/2 |
| Positive self | 2 | 1 | 1 | 1.33/10 |
| Human relationship | 4 | 5 | 4 | 4.33/4 |
| Civic consciousness | 1 | 2 | 1 | 1.33/9 |

[1] Evaluation scale from 1 to 10: 1—minimum grade (less important), 10—maximum grade (the most important).
Note: authors' elaboration.

**Table 5.** Impact of teaching method on competency development in the "Robotics" course (expert method).

| Competency | New Weight/Rank (Transferred from Previous Table) = a | Teaching Method 1 = VR = $b_1$ | Teaching Method 2 = Programming = $b_2$ | Teaching Method 3 = Testing = $b_3$ |
|---|---|---|---|---|
| Troubleshooting | 4.66 = $a_1$ | 5 [1] | 6 | 3 |
| Practical application | 4.66 = $a_2$ | 7 | 3 | 5 |
| Study map | 4.33 = $a_3$ | 4 | 3 | 3 |
| Human relationship | 4.33 = $a_4$ | 6 | 4 | 7 |
| Challenging practice | 4.33 = $a_5$ | 6 | 5 | 6 |
| Communication | 4.0 = $a_6$ | 3 | 4 | 2 |
| Interdisciplinary fusion | 3.66 = $a_7$ | 3 | 6 | 4 |
| Major (foundation) | 3.66 = $a_8$ | 6 | 6 | 5 |
| Weight of teaching method (W)/Rank [2] | | 170.14 | 153.82 | 147.5 |

[1] Evaluation scale is from 1 to 10: 1—minimum linkage between teaching method and competency development, 10—maximum linkage between teaching method and competency development. [2] Calculation method: W (weight of teaching method) = $(a_1 \times b_1) + (a_2 \times b_1)\ + \ldots (a_n \times b_1)$. Note: authors' elaboration.

According to the results in Table 6, we can see that the VR teaching method has the highest impact on competency development in the "Robotics" course because it develops the most important competencies better than the other teaching methods do.

### 3.2. Focus Group Interview (Professor/Student)

We chose the focus group interview method for data collection because it has the following advantages:

- It is a face-to-face method;
- It uses a moderator (the professor can hold this interview);
- It uses a fixed location and group (KOREATECH University, Mechanical Engineering Department, one classroom);
- Questions from an unexpected area (VR, for example) can be used;
- The period of the interview process is fixed (three times during one semester).

### 3.2.1. Participants

The participants were picked from the School of Mechanical Engineering (KOREATECH), and all of them were students of the second grade. We interviewed two groups of students: one group

was from the regular class of "Robotics" with a total of 62 participants (53 males and nine females); another group was from the VR class of "Robotics" with a total of 51 participants (39 males and 12 females). The interviewers were the Professor of the "Robotics" class and the manager of the human resources development department (HRD) of KOREATECH. They interviewed two "Robotics" classes: one regular class and one VR class. The developed questionnaire was the same for both groups of respondents. The structure of the interview process is presented in Table 6.

**Table 6.** Structure of interview.

| Regular Class "Robotics" | | | VR Class "Robotics" | | |
|---|---|---|---|---|---|
| Participants | Moderator | Period | Participants | Moderator | Period |
| 62 (53 + 9) | HRD | 1st stage: at the beginning of semester; 2nd stage: after mid-term exam (middle of semester); 3rd stage: at the end of semester (after final exam) | 51 (39 + 12) | Professor | 1st stage: at the beginning of semester; 2nd stage: after mid-term exam (middle of semester); 3rd stage: at the end of semester (after final exam) |

Note: authors' elaboration.

3.2.2. Interview

An example of the interview questions is provided in Appendix B. The interview was held three times during one semester: the first stage was at the beginning of the semester; the second stage was after the mid-term exam (middle of the semester); the third stage was at the end of the semester (after the final exam). There were 44 questions in total, and they were divided into eight different areas related to the competency model. Students had to answer only those questions which contributed to the evaluation of the most important competencies for the "Robotics" course. We introduced the students to the competencies and their meanings during the interview procedure.

Competency 1. Interdisciplinary fusion. All questions were related to the interdisciplinary correlation of this course with other courses. We ensured that the students understood the scope of the course.

Competency 2. Troubleshooting. All questions helped understand whether students were able to recognize the problem and find ways to solve it.

Competency 3. Communication. All questions focused on communication activities and cooperation within a group. Students presented their opinion on which type of decision-making process (individual or group) was more convenient for them.

Competency 4. Study map. All questions were based on course materials, teaching methods, and goals. Thus, we checked whether students understood everything, whether they could use the material, and which teaching method was the most attractive to them.

Competency 5. Challenging practice. These questions focused on academic and technical skills and understanding the project's trajectories.

Competency 6. Practical application. This pool of questions was related to the market area, practical skills, and self-development.

Competency 7. Human relationship. These questions helped check the student's understanding of the human features of a project in the robotics area.

Competency 8. Major foundation. This pool of questions focused on major knowledge and technical deep learning.

We used the following evaluation scale of five grades as possible answers:

(1)　−2 = very negative/totally disagree;
(2)　−1 = negative/disagree;
(3)　0 = not sure/so-so;
(4)　1 = good/agree;
(5)　2 = very good/exactly agree.

## 4. Results and Discussion

### 4.1. Results (Step 1 of Research)

Based on the expert method, we received the following summarized results:

1.  Syllabus analysis. Most of the topics focus on technical and communicative skills. Three main teaching methods are used: VR, programming, and testing. The course description is the same for the regular class and VR class; also, these two classes have similar goals.
2.  Competency identification. Eight main competencies were investigated (from a total of fourteen) that are necessary to develop in the "Robotics" course: interdisciplinary fusion, troubleshooting, communication, study map, challenging practice, practical application, human relationship, and major foundation. All these competencies had an average weight of more than 3.5 points.
3.  Teaching method evaluation. From analyzing the results, we found that virtual reality was the most effective for developing competencies. It ranked first with 170.14 points from the evaluation.
4.  Questionnaire formulation. On the basis of competency identification, we created 44 questions (around 5–7 for each competency) and provided a five-point scaled evaluation system for the answers.

### 4.2. Results (Step 2 of Research)

We performed the focus group interview in three stages, with the same participants interviewed throughout the semester. The results of the first interview at the beginning of the semester are shown in Table 7.

The survey of students during the first stage of the study was conducted at the beginning of the academic semester. At the time of the survey, the students were presented with the course and explained its goals, objectives, and methods.

Let us analyze the level of development of students' competencies at the first stage of the interview.

1.  Students' understanding of the "interdisciplinary fusion" competency in the two types of classes was extremely low; that is, they did not understand how this course related to other disciplines ("totally disagree" was chosen by 26% of the regular class and 25% of the VR class).
2.  We found a similar result in the two classes for the "troubleshooting" competency analysis; in the regular class, 33% of the students understood problem development (answered "agree"), and, in the VR class, 29% of the students expected to solve problems using virtual tools (answered "agree"). We suggest that VR application does not affect students' ability to investigate problems and formulate solutions at the beginning of the academic semester.
3.  "Communication" competency: students in the regular class did not expect a communication process in their class (30% of the respondents chose "disagree"); in the VR class, the results were the same—22% of the students were not sure about the communication process, but felt that it had some importance.
4.  All students understood the importance of this class (both the regular class (31% answered "exactly agree") and the VR class (25% answered "exactly agree")). This means that they adequately developed the competency "study map", with the regular class scoring a little bit better for this competency.
5.  The results for "challenging practice" competency were about equal between the two classes ("exactly agree" was the answer chosen by 27% of the regular class and 25% of the VR class). Students expected new knowledge and experience in this class regardless of teaching methods and materials.
6.  We found very different results for the two classes in the "practical application" competency analysis: in the regular class, 27% of the students did not expect practical study (answered "totally disagree"), and, in the VR class, 28% of the students expected a lot of practical studies using

virtual tools ("exactly agree"). We suggest that VR application has an influence on students' expectations for developing their practical skills.

7.  Approximately equal results were found in the "human relationship" competency analysis: in both classes, the students understood the importance of "Robotics" in human factor development and its impact on society ("agree" was answered by 30% of the regular class and 32% of the VR class).

8.  The "major foundation" competency had a more negative result in the regular class (35% of the students thought that this class did not provide major tools and approaches). The VR class had 24% negative answers. We suggest that VR application can motivate students to accept more major tasks and outcomes from the course.

Let us analyze the level of development of students' competencies at the second stage of the interview (Table 8).

1.  Students' understanding of the "interdisciplinary fusion" competency in the two types of classes was better than that at the beginning of the semester: 26% of the students in the regular class understood the interdisciplinary fusion of this subject, and 20% of the respondents in the VR class were not sure about interdisciplinary fusion. We suggest that VR does not improve this competency during the first half of the academic semester.

2.  The diverging result we found in the analysis of the "troubleshooting" competency remained: in the regular class, 28% of the students did not have expectations for problem development (negative answer), and, in the VR class, 32% of the students expected to solve problems using virtual tools ("exactly agree" answer). We suggest that VR application continues to motivate students to investigate problems and formulate solutions during the first half of the course's period.

3.  "Communication" competency: students in the regular class did not expect to engage in the communication process (30% of the respondents chose the negative answer); in the VR class, the results were better, as 20% of the students were not sure about the communication process but felt it had some importance.

4.  All students exactly understood the importance of this class (both the regular class (34% "exactly agree" answers) and the VR class (26% "exactly agree" answers)). This means that they adequately developed the "study map" competency.

5.  We got the same result for the two classes for the analysis of the "challenging practice" competency (29% "exactly agree" answers in the regular and VR classes). Students expected new knowledge and experience in this class regardless of teaching methods and materials.

6.  The diverging result we found in the "practical application" competency analysis remained: in the regular class, 25% of the students did not expect practical study (negative answer), and, in the VR class, 32% of the students expected a lot of practical studies using virtual tools ("exactly agree" answer). We suggest that VR application influences students' expectations for developing their practical skills.

7.  Approximately equal results for the two classes were found in the "human relationship" competency analysis: in both classes, students understood the importance of "Robotics" for human factor development and its impact on society ("agree" was answered by 33% of the regular class and 25% of the VR class).

8.  The "major foundation" competency had different results in the two different classes. There was an overall negative result in the regular class (26% of the students thought that this class did not provide major tools and approaches). The VR class had 24% positive answers. We suggest that VR application can stimulate students to accept more major tasks and outcomes from the course.

Let us analyze the level of development of students' competencies at the last stage of the interview (Table 9).

**Table 7.** Comparative results of first-stage interview (regular class and VR class).

| Competency | Regular Class Results, % | | | | | VR Class Results, % | | | | |
|---|---|---|---|---|---|---|---|---|---|---|
| | −2 (Very Negative/ Totally Disagree) | −1 (Negative/ Disagree) | 0 (Not Sure/ So-So) | 1 (Good/ Agree) | 2 (Very Good/ Exactly Agree) | −2 (Very Negative/ Totally Disagree) | −1 (Negative/ Disagree) | 0 (Not Sure/ So-So) | 1 (Good/ Agree) | 2 (Very Good/ Exactly Agree) |
| 1. Interdisciplinary fusion | 26% | 27% | 12% | 23% | 11% | 25% | 21% | 7% | 17% | 13% |
| 2. Troubleshooting | 14% | 15% | 8% | 33% | 31% | 10% | 10% | 7% | 29% | 26% |
| 3. Communication | 23% | 30% | 14% | 15% | 18% | 19% | 22% | 11% | 16% | 15% |
| 4. Study map | 15% | 15% | 9% | 31% | 30% | 13% | 12% | 11% | 25% | 22% |
| 5. Challenging practice | 21% | 22% | 7% | 23% | 27% | 18% | 15% | 5% | 19% | 25% |
| 6. Practical application | 27% | 20% | 11% | 21% | 22% | 11% | 8% | 6% | 29% | 28% |
| 7. Human relationship | 28% | 21% | 4% | 30% | 17% | 7% | 9% | 13% | 32% | 21% |
| 8. Major foundation | 35% | 35% | 10% | 10% | 11% | 23% | 24% | 9% | 15% | 12% |

Note: authors' elaboration.

**Table 8.** Comparative results of second-stage interview (regular class and VR class).

| Competency | Regular Class Results, % | | | | | VR Class Results, % | | | | |
|---|---|---|---|---|---|---|---|---|---|---|
| | −2 (Very Negative/ Totally Disagree) | −1 (Negative/ Disagree) | 0 (Not Sure/ So-So) | 1 (Good/ Agree) | 2 (Very Good/ Exactly Agree) | −2 (Very Negative/ Totally Disagree) | −1 (Negative/ Disagree) | 0 (Not Sure/ So-So) | 1 (Good/ Agree) | 2 (Very Good/ Exactly Agree) |
| 1. Interdisciplinary fusion | 22% | 20% | 8% | 26% | 24% | 17% | 10% | 20% | 16% | 19% |
| 2. Troubleshooting | 23% | 28% | 5% | 20% | 23% | 7% | 7% | 5% | 31% | 32% |
| 3. Communication | 23% | 30% | 14% | 15% | 18% | 14% | 20% | 20% | 15% | 14% |
| 4. Study map | 13% | 13% | 9% | 31% | 34% | 10% | 13% | 9% | 24% | 26% |
| 5. Challenging practice | 18% | 19% | 10% | 24% | 29% | 13% | 11% | 3% | 26% | 29% |
| 6. Practical application | 25% | 18% | 11% | 24% | 22% | 6% | 7% | 6% | 31% | 32% |
| 7. Human relationship | 24% | 17% | 4% | 33% | 22% | 7% | 9% | 17% | 25% | 25% |
| 8. Major foundation | 23% | 26% | 7% | 24% | 20% | 13% | 17% | 8% | 24% | 20% |

Note: authors' elaboration.

**Table 9.** Comparative results of third-stage interview (regular class and VR class).

| Competency | Regular Class Results, % | | | | | VR Class Results, % | | | | |
|---|---|---|---|---|---|---|---|---|---|---|
| | −2 (Very Negative/ Totally Disagree) | −1 (Negative/ Disagree) | 0 (Not Sure/ So-So) | 1 (Good/ Agree) | 2 (Very Good/ Exactly Agree) | −2 (Very Negative/ Totally Disagree) | −1 (Negative/ Disagree) | 0 (Not Sure/ So-So) | 1 (Good/ Agree) | 2 (Very Good/ Exactly Agree) |
| 1. Interdisciplinary fusion | 15% | 18% | 11% | 26% | 30% | 12% | 8% | 12% | 23% | 27% |
| 2. Troubleshooting | 26% | 25% | 11% | 17% | 21% | 5% | 5% | 5% | 28% | 39% |
| 3. Communication | 12% | 22% | 22% | 25% | 19% | 13% | 18% | 16% | 19% | 16% |
| 4. Study map | 12% | 10% | 9% | 31% | 38% | 8% | 10% | 6% | 30% | 29% |
| 5. Challenging practice | 17% | 16% | 13% | 29% | 25% | 8% | 6% | 3% | 29% | 36% |
| 6. Practical application | 24% | 15% | 11% | 26% | 24% | 8% | 9% | 5% | 32% | 29% |
| 7. Human relationship | 23% | 17% | 10% | 29% | 21% | 5% | 9% | 13% | 29% | 27% |
| 8. Major foundation | 21% | 24% | 8% | 25% | 22% | 9% | 14% | 10% | 24% | 25% |

Note: authors' elaboration.

1.　Students' understanding of the "interdisciplinary fusion" competency in the two types of classes was much better than that at the beginning of the semester: 30% of the students in the regular class understood the interdisciplinary fusion of this subject, and 27% of the respondents in the VR class were positive about interdisciplinary fusion. We saw that VR improved this competency during the academic semester.

2.　We found a diverging result in the analysis of the "troubleshooting" competency: in the regular class, 26% of the students did not expect problem-solving skill development ("totally disagree" answer), and, in the VR class, 39% of the students expected to solve problems using virtual tools ("exactly agree" answer). We suggest that VR application increasingly motivated students to investigate problems and formulate solutions throughout the whole course period.

3.　For the "communication" competency, students in the regular class changed their minds between the second and third stages, and they expected to engage in the communication process (25% of the respondents "agree"); in the VR class, the effect was less, as 19% of the students supported the communication process and understood the importance of it.

4.　All students exactly understood the importance of this class (both the regular class (38% "exactly agree" answers) and the VR class (30% "exactly agree" answers). This means that they adequately developed the "study map" competency at the beginning and the end of the academic semester.

5.　We got equal results for the two classes in the analysis of the "challenging practice" competency ("agree" was answered by 29% of the regular class, and "exactly agree" was answered by 36% of the VR class). Students expected new knowledge and experience in this class at the beginning of the semester regardless of teaching methods and materials; however, during the course period, the expectations in the VR class increased.

6.　We found a similar result in the "practical application" competency analysis: in the regular class, 26% of the students expected practical study ("agree" answer) at the end of the semester, and, in the VR class, 32% of the students expected a lot of practical studies using virtual tools ("agree" answer). We suggest that VR application does not influence students' expectations for developing their practical skills at the end of course.

7.　Exactly equal results between the two classes were found in the "human relationship" competency analysis: in both classes, the students understood the importance of "Robotics" in human factor development and its impact on society (29% "agree" answers in both classes).

8.　The "major foundation" competency had similar results in both classes. There were positive results in both classes (25% of the students thought that this class provided major tools and approaches).

Now, let us compare the overall results among the three stages (first, second, and third interviews) (Table 10). We submit the most representative answers (those with the highest percentage) in each column. The minus sign indicates a negative answer, and the plus sign indicates a positive answer.

**Table 10.** Comparative study of three interview stages (regular class and VR class).

| Competency | Regular Class Results, % | | | VR Class Results, % | | |
|---|---|---|---|---|---|---|
| | 1st Interview | 2nd Interview | 3rd Interview | 1st Interview | 2nd Interview | 3rd Interview |
| 1. Interdisciplinary fusion | −27% | +26% | +30% | −25% | 20% | +27% |
| 2. Troubleshooting | +33% | −28% | −26% | +29% | +32% | +39% |
| 3. Communication | −30% | −30% | +25% | −22% | 20% | +19% |
| 4. Study map | +31% | +34% | +38% | +25% | +26% | +30% |
| 5. Challenging practice | +27% | +29% | +29% | +25% | +29% | +36% |
| 6. Practical application | −27% | −25% | +26% | +29% | +32% | +32% |
| 7. Human relationship | +30% | +33% | +29% | +32% | +25% | +29% |
| 8. Major foundation | −35% | −26% | +25% | −24% | +24% | +25% |

Note: authors' elaboration.

We compared the results of the two classes from the three stages of the interview process and understood that different competencies could be developed with different methods of education. However, four competencies ("interdisciplinary fusion", "communication", "challenging practice", and "major foundation") were highly developed in the VR class, and the results changed from negative answers to positive answers during the survey. We suggest that VR application has a more positive effect on these competencies than other teaching tools and materials.

Therefore, we can accept and reject the following hypotheses:

1.  **Hypothesis 1**, *"The VR class improves and develops core competencies in a basic engineering course better than in the regular class"*, *was not proven through this study*.
2.  **Hypothesis 2,** *"The VR class does not improve or develop core competencies in a basic engineering course better than in the regular class"*, *was not proven through this study*.
3.  **Hypothesis 3,** *"The VR class improves and develops only a certain scope of competencies in a basic engineering course"*, *was proven through this study*.

*4.3. Discussion*

Most modern teaching methods (flip-learning, problem-based learning, project-based learning) can be used together with VR/AR applications. Gaunet et al. and Schuh et al. [55–57] indicated that VR application helps users recognize problems in a given environment and leads to students investigating many solutions using simulations and experiments. These are some of the benefits of VR application in engineering education.

Also, some disadvantages of VR application in engineering education should be recognized, and some discussions are provided by Mosconi, Buguin et al., and Schwab [58–62]: VR implementation is very costly, and it presents difficulty during an experiment's development, team-building, and identifying the learner's role.

Through this research, we investigated the impact of different teaching methods on students' competency development; the effect of VR training in class was analyzed. The CQI (channel quality indication) report is an important element of the teaching process in Korean Universities. It has a significant impact on the professor's teaching system performance because students complete class evaluations at the end of the semester. There are two types of CQI reports: periodic (for general classes) and aperiodic (for special cases, for example, short lectures). In our survey, we refer to the periodic CQI report. We still have an open question: are the results of the focus group interview correlated with the results of the CQI report at the end of the academic semester? This means that we should investigate the impact of competency development in the VR class on the results from the CQI report. In this case, we can achieve more specific results by using data from different trajectories of the survey (students' interviews; CQI report analysis; National Competency Standards). The correlation between competency development and the CQI report is presented in Figure 4.

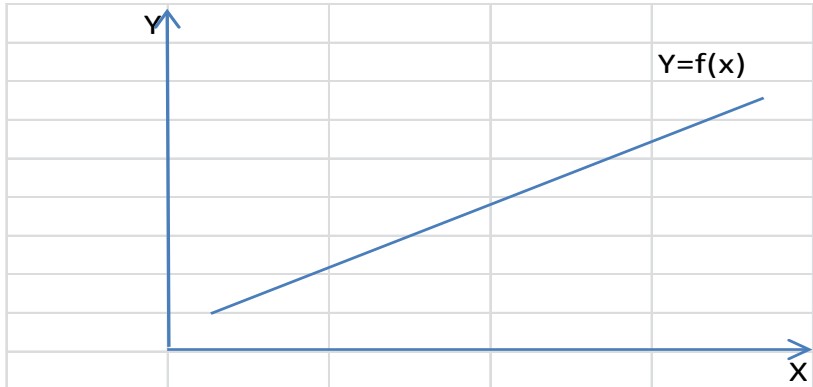

**Figure 4.** Correlation between competency development and channel quality indication (CQI) report. Note: authors' elaboration. X—level of competency development; Y—grade in the CQI report.

For future research, it is necessary to investigate the correlation between VR application and online education in the engineering field.

The online classroom is undoubtedly the educational segment that can grow the most using VR technologies. Market trends indicate that distance learning through the internet grows in double-digits in all markets. This leads to an increase in the number of users and, thus, to an increase in the supply in response to existing potential business. Increased competition, following the logic of business, will increase suppliers' quality of products to capture a greater share of the market, since price reduction has a lower limit that lies in the transition from prestige to disrepute of the title to be obtained. Rüßmann et al., and Rodic, Sommer, and Brettel [63–65] reported that, when products are too cheap in the education sector, they no longer fulfill their role in the professional market because they are not accepted as valid education when seeking to obtain a job.

Shvetsova O. [66] asserted that traditional online teaching is at the limit of service exploitation through current platforms: inclusion in social networks, collaborative learning through conversations, video and audio, virtual tutorials using videoconferencing systems, links to forums, multichannel access to information databases around the globe, etc. The platforms do not have more capacity. Perhaps it is time to focus on improving the experience within the teaching field itself, i.e., improving what is currently available to the student: texts, videos, and problems to be solved individually or in a group (but always dependent on the imagination and the creative capacity of the end user). This will no doubt change with the integration of educational experiences through the use of virtual and augmented reality, as reported by Gerlitz and Sanders [67,68].

From the conceptual point of view or the point of view of service, virtual and augmented reality through online channels will be realized in the future because educational programs introduce real experiences by making use of these technologies. Allmada-Lobo [69] predicted that this will further equalize distance learning with face-to-face teaching, making use of the flexibility that, by definition, distance learning already entails. The new virtual reality systems applied to collaborative social systems will allow students to attend a class and perceive that they are sitting in the classroom surrounded by their classmates and seeing the teacher in front of them teaching the subject, as suggested by Maier [70]. There are already simple applications that allow us to envision this concept as being tangible very soon. Classrooms will be located anywhere on the planet, leaving behind the feeling of solitude that a webpage or an online learning platform confers. Müller et al. [71] confirmed that the goal and, with it, the future of online education will be to get the best of both worlds: the benefits offered by the flexibility of access without preset hours and the convenience of learning from home, along with the advantages of accessing experiences, equipment, companions, and professors that provide online students with the same educational value as face-to-face teaching.

## 5. Conclusions

This research focused on competency development in an engineering class that used a VR application. The study was conducted on the basis of the Fourth Industrial Revolution and the Korean National Competency Standards. The aim of this research was to conduct a comparative study of two university engineering courses to present the difference between the effects of VR-based teaching and traditional teaching on learning robotics. To achieve this goal, a complex survey was developed that included eight steps. The authors developed three hypotheses and four research questions that proposed the relationship between the teaching method, competency development, and VR effects, some of the concepts were developed by authors earlier. The survey was based on a case study of Korea University of Technology and Education. The authors used several research methods, namely, the expert method, focus group interview, and comparative study. The participants were second- and third-grade university students. The quantitative data were collected through the interview forms developed by the researchers; the qualitative data were analyzed through the expert method and a comparative study. A significant difference was observed between competency development in a VR class and a regular class with a similar syllabus. Moreover, it was noted that different teaching

methods affect competency development in different ways, and VR tools develop some competencies, but not all of them.

The main results of this research are as follows:

1.  Different teaching methods have different effects on competency development in engineering education;
2.  Hypothesis 1, "The VR class improves and develops core competencies in a basic engineering course better than in the regular class", was not proven through this study.
3.  Hypothesis 2, "The VR class does not improve or develop core competencies in a basic engineering course better than in the regular class", was not proven through this study.
4.  Hypothesis 3, "The VR class improves and develops only a certain scope of competencies in a basic engineering course", was proven through this study.
5.  There is a more positive tendency to develop competencies in the VR class than in the regular class;
6.  Four competencies ("interdisciplinary fusion", "communication", "challenging practice", and "major foundation") were highly developed in the VR class, and the results changed from negative answers to positive answers during the semester, according to the survey. We suggest that VR application can affect these competencies more than other teaching tools and materials.
7.  VR stimulated students to develop their communication and team-building skills and motivated them to more actively develop their practical skills.

For future research, it is necessary to investigate the correlation between VR application and online education in the engineering field.

**Author Contributions:** J.H.L. and O.A.S. conceptualized the study; J.H.L. designed the methodology; O.A.S. carried out the investigation; J.H.L. and O.A.S. did the analyses, validation, and data curation; O.A.S. wrote the paper, and J.H.L. contributed to reviewing and editing all sections; J.H.L. supervised the work.

**Funding:** This paper received research funding in 2018 from Korea University of Technology and Education (KOREATECH).

**Conflicts of Interest:** The authors declare no conflict of interest.

## Appendix A

SYLLABUS "Robotics" (VR/Regular class)

Lectures: two sessions/week, 1.5 h/session

Labs: one session/week, two hours/session

This course is a restricted elective (professional subject).

Main reference: Asada, H. and Slotine, J. J. *Robot Analysis and Control.* New York, NY: Wiley, 1986. ISBN: 9780471830290.

Labs and Projects:

Lab sessions are geared toward completion of two projects: a de-mining robot and a robot that finds and rescues disaster victims.

**Table A1.** Grading policy.

| Activities | Percentages |
| --- | --- |
| Mid-term exam | 30% |
| Final exam | 30% |
| Homework | 20% |
| Laboratory and design project | 20% |

The calendar below provides information on the course's lecture (L) and lab (Lab) sessions.

**Table A2.** Calendar.

| # | Topics | Key Dates |
|---|--------|-----------|
| L1 | Introduction | |
| L2 | Actuators and drives | |
| L3 | Control components | Problem set 1 out |
| Lab 1 | De-mining robot: embedded robot controller, intelligence interface, and power machine amplifiers | |
| L4 | Control software 1 | Problem set 1 due |
| Lab 2 | De-mining robot: controller software and sensor inputs | |
| L5 | Control software 2 | Problem set 2 out |
| L6 | Sensors 1 | |
| Lab 3 | De-mining robot: implement basic sensor-based controls; plan strategy for de-mining task | |
| L7 | Kinematics 1 | Problem set 3 out |
| L8 | Kinematics 2 | Problem set 2 due |
| Lab 4 | De-mining robot: refine de-mining operations | |
| L9 | Differential motion 1 | Problem set 3 due Problem set 4 out |
| Lab 5 | Rescue robot: stage A—concept design | |
| L10 | Differential motion 2 | |
| L11 | Statics, energy method | Problem set 4 due Problem set 5 out |
| Lab 6 | Rescue robot: stage B—prototype implementation | |
| L12 | Hybrid position-force control | |
| L13 | Compliance, end-effecter design | Problem set 5 due |
| Lab 7 | Rescue robot: stage B—prototype implementation (cont.) | |
| L14 | Non-holonomic systems | Problem set 6 out |
| L15 | Mid-term exam | |
| Lab 8 | Rescue robot: stage C—system integration | |
| L16 | Legged robots, multi-fingered hands | |
| L17 | Dynamics 1 | Problem set 6 due Problem set 7 out |
| L18 | Dynamics 2 | |
| L19 | Computed torque control | Problems set 7 due |
| Lab 9 | Rescue robot: stage C—system integration (cont.) | |
| L20 | Sensors 2 | Problems set 8 out |
| L21 | Computer vision | |
| L22 | Navigation 1 | |
| L23 | Navigation 2 | Problems set 8 due |
| Lab 10 | Rescue robot: stage D—testing | |
| L24 | Tele-robotics and virtual reality | |
| L25 | Final exam | |
| L26 | Check-out of final projects | |
| L27 | Project demonstration | |

## Appendix B

**Table A3.** Interview list.

| Competency | Statement/Question | Evaluation scale | | | | |
|---|---|---|---|---|---|---|
| | | −2 (Very Negative/ Totally Disagree) | −1 (Negative/ Disagree) | 0 (Not Sure/ So-So) | 1 (Good/ Agree) | 2 (Very Good/ Exactly Agree) |
| 1. Interdisciplinary fusion | I understand how this class correlates with other subjects | | | | | |
| | I can use class tools in other areas | | | | | |
| | I think that I can meet the same teaching methods in other classes | | | | | |
| 2.Troubleshooting | I can understand all problems and tasks | | | | | |
| | I can make my own decision for problem-solving | | | | | |
| | Teaching methods help me to investigate problems | | | | | |
| | I have critical thinking skills | | | | | |
| | It is more effective to solve problems in group | | | | | |
| | In this class, it is better to solve problems individually | | | | | |
| 3. Commun-ication | I can communicate easily with classmates and the professor | | | | | |
| | I reduce the time of communication | | | | | |
| | I improve my communication skills within this class | | | | | |
| | I can introduce a team-project presentation | | | | | |
| | I need more channels for communication | | | | | |
| 4. Study map | I understand the goals of this class | | | | | |
| | I like the teaching methods and understand the material | | | | | |
| | An individual task is better than a group task | | | | | |
| | VR application helps me in class (I think/I suggest) | | | | | |
| | I need more visualization in this class | | | | | |
| 5. Challenging practice | I get new knowledge in this subject | | | | | |
| | I understand how to use this knowledge in the market | | | | | |
| | I know how to share skills and knowledge of this class | | | | | |
| | I may conduct technical research | | | | | |
| | I have academic understanding of robotics | | | | | |
| | I have technical understanding of robotics | | | | | |
| | I think that I can choose robotics for my graduate thesis | | | | | |
| 6. Practical application | I can create my own project in this class | | | | | |
| | I can participate in group activity | | | | | |
| | I do a lot of practical tasks and improve my skills | | | | | |
| | I understand all visualization aspects of robotics | | | | | |
| | I want to participate in research and development (R&D) with my project | | | | | |
| | I want to present my project in conference (other event) | | | | | |
| 7. Human relationship | I know in which areas I can use robotics | | | | | |
| | I understand perspectives of robot application in human industries | | | | | |
| | I know a lot of practical tools for how to create robots for life improvement | | | | | |
| | I understand safety considerations for robot operations | | | | | |
| 8. Major foundation | I understand how to plan a robot trajectory | | | | | |
| | I understand the robot dynamics | | | | | |
| | I know how to improve kinematics, data parameters in robotics | | | | | |
| | I understand sensors and actuators | | | | | |
| | I understand homogeneous transformation | | | | | |
| | I can present initial and final postures in robot planning | | | | | |
| | I understand how to implement the trajectory planning in the real robot and verify it | | | | | |
| | I know implementation and robot economics | | | | | |

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
