# Peer review of "The Impact of VR Application on Student’s Competency Development: A Comparative Study of Regular and VR Engineering Classes with Similar Competency Scopes"

_sustainability, doi:10.3390/su11082221_

Round 1

Reviewer 1 Report

l   The research findings are not valuable/significant.

A significant difference was observed between competencies’ development in VR class and regular class with similar Syllabus. Besides, it was noted that different teaching methods affect competencies development in different way, and VR tools develop not all competencies, but some of them.

l   There is no explanation on why this research is required in the Introduction section.

l   More exact citations should be considered. For example, the information on lines from 99-107 seems missing.

l   Why is the Table 1 necessary?

l   Where and how to collect the information in Table 3?

l   There are two section numbered as 2.

l   There is no explanation on Figures 2 and 3.

l   What’s the expert method?

l   The authors designed the evaluation scale from 1 to10, however, the respondents just selected 1-5. Why?

l   What’s CQI?

Author Response

Dear reviewer! Please find our respond as attached file. Thank you for cooperation!

Reviewer 2 Report

Authors investigated the application of Virtual Reality in education field, which is explored by many researchers in their own way. I have the following comments based on my understanding:

- Line 39, reference 1-2 are not related with the topic as mentioned by the authors.

- Line 39, reference number 3-5 are not relevant to the mentioned sentence, I checked these articles, and none of them discusses about VR.

- Line 45, " Virtual reality (VR) is a model of reality created....." once you defined the acronym in the first paragraph then no need for repetition

- Line 50, "Reference 7-10 are wrongly cited , they are not explaining any concept of VR as mentioned by authors.

- Line 89, "The most important question is how VR application affects education process in different areas. Most of researchers (Hameed, I. at al.; Bandura, A.; Diseth, A.) tell us about advantages of VR technologies in engineering, medicine and earth science field [12-14]..... Same issue, the mentioned articles even do not mention the term VR".

I will not able to read the further sections, since I think that the article was written in a careless manner. I hope authors will improve these mistakes.

Author Response

Dear Reviewer! Please, find our response as attached file. Thank you for cooperation!

Reviewer 3 Report

It's an interesting paper that's addresses a preeminent question of how virtual reality fits in both the evolution of learning in the context of the 4th industrial Revolution.

Several initiatives of application of virtual reality to different situations and to teaching have been developed but few investigate how effective it is in a quantitative way and differentiating which skills and competences are affected.

Title is adequate.

The major flaw off the paper it's the quality of writing. The text needs a major revision to bring it up to the standard of an international journal. Although the text is thoroughly commented and annotated, many errors may remain.

 Content

Given that appraisal of virtual reality in the learning process is the aim of the paper, the VR application should be thoroughly described. After reading the paper the reader has no idea if the students watched classes in virtual reality, watched animations, videos or used interactive development environments.

In section 2.2 there is a complete discussion of Industry 4.0 but little of VR in this context, and it is misplaced, as indicated on the file.

Author Response

Dear reviewer! Please, find our response as attached file. Thank you for cooperation!

Round 2

Reviewer 1 Report

1. The illustration about research outcomes in the abstract is limited.

2. Section 1 is too long. It is suggested to move some of the contents into another section.

3. Some references might be lost. For example, "Such changes are already being implemented in leading foreign universities and can be classified as follows:" is it cited from any reference?

4. One researcher’s perspective on VR application in the global environment. Who?

5. This sentence should not be an independent paragraph: Electronic (digital, web, internet) economics is defined as economic activity based on digital technology. This is a basic writing rule.

6. What's the research problem? It should be highlighted and the research questions can be linked. Why do we require the difference (illustrated in the research goal)? Please illustrate the relationship between research problem(s) and research goal.

7. Why to put the last paragraph of 2.2 there? It is strange.

8. Subsection 2.4 is not complete as an independent subsection.

9. What is the difference of Major (foundation) and Major (specialization), and how to evaluate?

10. In subsection 4.2, the authors illustrate the hypotheses were proven or not. How to confirm those? The evidence is weak.

Author Response

Dear reviewer!

thank you very much for your recommendations and feedback. We hope that we met all your requirements and recommendations in our manuscript,

Kind regards, Authors

Reviewer 2 Report

The changes improved the quality of manuscript now.

Author Response

(The authors gave the same response as above.)

Round 3

Reviewer 1 Report

All comments have been addressed.